# Cross-border climate vulnerabilities of the European Union to drought

Ertug Ercin ⓘ [1,2 ✉], Ted I. E. Veldkamp ⓘ [3] & Johannes Hunink[4]

European Union's vulnerability to climate change stretches far beyond its borders because many of its economic sectors, such as meat and dairy, use raw materials sourced from far afield. Cross-border climate vulnerability is a relatively new subject in scientific literature, while of high societal and economic relevance. We quantify these climate vulnerabilities with a focus on drought risk and assessed them for 2030, 2050, 2085 and for RCP 2.6 and 6.0 climate scenarios. Here we find that more than 44% of the EU agricultural imports will become highly vulnerable to drought in future because of climate change. The drought severity in production locations of the agricultural imports in 2050 will increase by 35% compared to current levels of drought severity. This is particularly valid for imports that originate from Brazil, Indonesia, Vietnam, Thailand, India and Turkey. At the same time, imports from Russia, Nigeria, Peru, Ecuador, Uganda and Kenya will be less vulnerable in future. We also report that the climate vulnerabilities of meat and dairy, chocolate (cocoa), coffee, palm oil-based food and cosmetic sectors mainly lie outside the EU borders rather than inside.

[1] R2Water Research and Consultancy, Amsterdam, the Netherlands. [2] Institute for Environmental Studies, Vrije Universiteit Amsterdam, Amsterdam, the Netherlands. [3] Amsterdam University of Applied Science, Amsterdam, the Netherlands. [4] FutureWater, Cartagena, Spain. ✉email: ercin@r2water.nl

Recent years have seen a rise in heat waves and unprecedented drought conditions in Europe. This has disrupted Europe's agricultural production[1–4]. According to climate change forecasts, such extreme weather events are likely to increase[5–7]. Yet Europe's vulnerability to extreme weather events and climate change stretches far beyond its borders because many of its economic sectors and food consumption, such as meat and dairy, use raw materials sourced from far afield[8]. This product flow through international trade means that these sectors are vulnerable to extreme weather events and climate change in the original production regions. For example, the European Union (EU) relies almost entirely on imports of soybean to meet the demand for animal feed rather than the use of locally grown crops[9]. The EU imports around 30–35 million tonnes of soybean per year and produce only 0.9 million tonnes per year domestically[10]. The deficit in soybean production in the EU poses a significant risk to its economy, especially to its meat and dairy industry since it is the main source of feed for animal husbandry[11–13]. This makes the EU highly vulnerable to any disruption of soybean production that may occur as a result of weather shocks, such as extreme heat, in the countries that produce soybeans for export to the EU. Consequently, droughts and lack of rainfall within the EU are not the only phenomena that could negatively affect its agricultural industry. Should drought occur in the regions that produce the food imported by the EU, it would disrupt supply. As a result, commodity prices would change, which could lead to economic damage and social disruption within the EU[14–16].

Over the past decade, a growing number of assessments have emerged from the scientific literature that focuses on the vulnerability of various sectors, including agriculture, to climate change and extreme weather events[17–21]. These studies have adopted the well-established Intergovernmental Panel on Climate Change (IPCC) definition of vulnerability, described as the extent to which a natural or social system is susceptible to sustaining damage from climate change impacts, and is a function of exposure, sensitivity, and adaptive capacity[22]. Most of these studies applied the vulnerability to application to a specific sector (e.g., coffee, agriculture) in a particular region and provided integrated assessments, in which several exposure and sensitivity components were merged into a single vulnerability framework. Climate vulnerability assessments of agriculture, meanwhile, focus on the production node within a geographic area, instead of analyzing it from both external and internal perspectives[23].

Consequently, knowledge and research on the cross-border climate vulnerabilities of a geographic region has been neglected and is still a relatively new topic in scientific literature[24–26]. Following the IPCC's Fifth Assessment Report (AR5) in 2014, which explored 'cross-regional phenomena'[27], some studies addressed cross-border climate vulnerabilities at a global, regional, and national scale[8,28–34]. They mainly focused on qualitative analysis, and on country-specific case studies, providing an insight into the possible climate impacts in hotspot exporting areas. As such, they lacked quantitative analyses and did not assess which crops and crop groups are most vulnerable or how vulnerabilities will change in the future, compared to current climatic conditions.

To bridge this knowledge gap, we quantified and mapped cross-border climate vulnerabilities of the EU's agri-food economy in relation to drought severity in third countries, expressed in cross-border climate vulnerability score (CCVS). To quantify and map the cross-border vulnerabilities, we first calculated the dependency of the EU's agri-food imports on the rainfall use at production locations, green virtual water imports[35], by the target years 2030, 2050, and 2085 using Shared Socioeconomic Pathways 2 (SSP2)[36] characteristics. Then, we selected key-imported crops based on their importance in the EU external rainfall dependency. In the next step, we calculated how drought severity changes in exporting locations under different climate change scenarios (under Representative Concentration Pathway 2.6 and 6.0). The last step is to assess the adaptive capacity of the exporting regions to climate change. Combining all these three elements provides cross-border climate vulnerabilities of the EU's agri-food economy to drought, expressed in terms of a score calculated as the multiplication of changes in exposure and sensitivity under different climatic conditions with adaptive capacity to climate change of the exporting regions.

## Results

**Cross-border climate vulnerabilities of the EU to drought.** Global climate change will make the EU's agri-food economy more vulnerable to drought in non-EU countries in the future, as seen in Fig. 1. The cross-border climate vulnerability score of the EU's agri-food economy to drought (CCVS), for 2030, 2050, and 2085 under RCP 2.6 and RCP 6.0 concentration pathways, falls between 1.20 and 1.35, which represents a moderate level of climate vulnerability. A CCVS of 1.25–1.35 means that the total

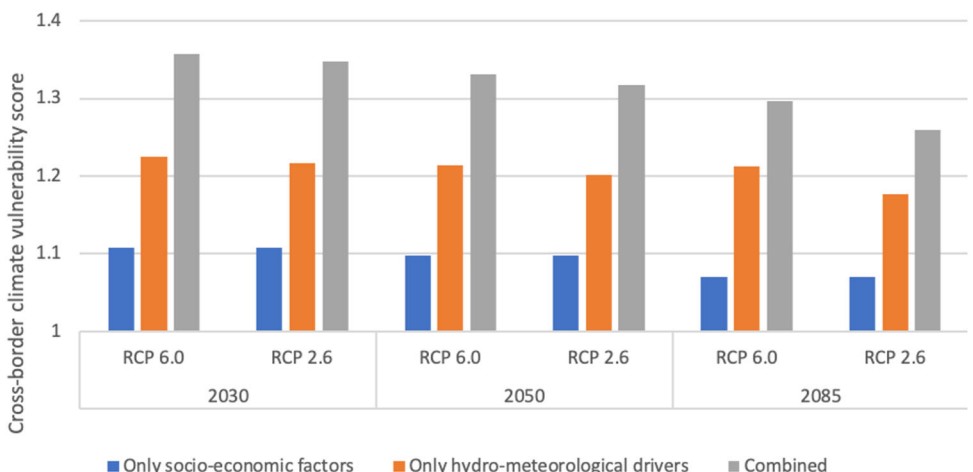

**Fig. 1 Cross-border climate vulnerability score (CCVS) of the EU's agri-food economy to drought for 2030, 2050, and 2085 under RCP 2.6 and RCP 6.0 concentration pathways.** The blue bars show CCVSs when only socio-economic drivers are taken into account; the orange bars when only hydro-meteorological drivers are taken into account; and the gray bars when both drivers are taken into account.

amount of agricultural imports by the EU will be 25–35% more vulnerable to drought in the future compared to the current situation mainly because of change in drought severity, increased intensity, and duration of drought events, in the production locations of the imported products.

Figure 1 also includes CCVSs disaggregated by type of drivers. We used two types of drivers in the calculations: (i) socio-economic drivers, such as population, Gross Domestic Product, trade policies, food demand, etc. (considering any consequent changes in the number of agricultural imports by the EU); and (ii) hydro-meteorological drivers for drought severity. By examining each of the two drivers separately, we find that changes in the severity of drought in exporting countries are the major determining factor behind the CCVSs. Figure 1 shows that CCVSs of the EU will peak (CCVSs at around 1.35) around 2030 for both RCP scenarios. They will remain at similar levels in 2050 (slightly decreased), before starting to go down in 2085. The decline observed is mainly related to population decrease in the EU (and consequent demand changes for agricultural imports by the EU) in 2085, compared to 2030 and 2050 population forecasts. Our analysis shows that the EU's agricultural imports will be less vulnerable to drought under the RCP 2.6 than the RCP 6.0 concentration pathway. The major difference is observed after 2050 and by 2085. This is consistent with the characteristics of RCP 2.6, which is a peak and decline scenario.

Although the CCVS of the EU's agri-food economy is more than 1.25, its spatial distribution across exporting countries differs significantly. For example, under the RCP 6.0 concentration pathway in 2050, CCVS of the EU's related to agricultural imports from Brazil (the largest green virtual water exporting country to the EU) is over 1.5, which indicates a high vulnerability level. The CCVS related to agricultural imports from other exporting countries is highest in Indonesia at over 3.5, which indicates a very-high vulnerability level for the EU imports. This is followed by India at 1.5. CCVSs related to other countries fall between 1 and 1.5, such as Ivory Coast, Ghana, Malaysia, Paraguay, Cameroon, and Argentina. In contrast, imports from some countries show a reduced vulnerability to drought in the future. For example, CCVSs related to agricultural imports from Nigeria, China, Ecuador, Peru, and Uganda are lower than 1 under the RCP 6.0 concentration pathway in 2050 (Fig. 2, map related to RCP 2.6 concentration pathway is provided in the supplementary information section, please see Fig. S1).

Adaptive capacity per exporting region can be an important element in assessing CCVS per exporting country. In some countries, the high adaptive capacity of agriculture to climate change (expressed in terms of equipped agriculture areas with irrigation, fertilizer and pesticide use, and tractor use, see the "Methods" section) reduces cross-border vulnerabilities. For example, the CCVS score of the United States of America (USA) without adaptive capacity component will be one, which is 0.2 higher than the CCVS with the adaptive capacity component. Similarly, a considerable reduction in CCVSs is observed in Malaysia (from 1.35 to 0.7) and Indonesia (from 3.9 to 3.5). In other large green water importing countries effect of adaptive capacity is negligible.

**Drought severity level of the EU's agricultural imports.** CCVSs tells us changes in vulnerabilities to drought under climate change compared to current, however, they do not reveal how severe the drought will be in exporting locations. To answer this question, we overlaid the volume of agricultural imports by the EU with drought severity maps under different climatic conditions. We used five drought severity levels at production locations: low; low-medium; medium; high; and extremely high[37]. Low drought

severity means that drought events are either short in time or affect a small spatial spread or both. High drought severity implies longer more frequent and wider-spread drought events.

Under the current climate around 93% of the agricultural imports to the EU come from locations with a low/low-medium drought severity. The rest (7%) are categorized as medium-high and high. This alters significantly under climate change, in 2050 under the RCP 6.0 concentration pathway, only 18% of the EU's agricultural imports come from locations with low drought severity and around 44% of the imports come from areas that will experience high and extremely high drought severity (Fig. 3).

**Cross-border climate vulnerability of the EU: key-imported crops.** The CCVSs also vary per crop (Fig. 4). This variation was identified by assessing the climate vulnerability of eight key crops that are imported to the EU separately: soybean, cocoa, coffee, oil palm, sunflower, maize, olives, and sugarcane. Of the key imported crops, sunflower and maize imports by the EU have the lowest CCVSs between 1.13 and 1.16 in 2050 under the RCP 6.0 concentration pathway, respectively. CCVSs related to the imports of three crops, cocoa, sugarcane, and palm oil, show high climate vulnerability to drought. They score over 2.0 for each RCP scenario in all target years (except cocoa for 2085). In addition, imports of olives and coffee are highly vulnerable to drought in the future (Fig. 4).

In Fig. 5, we present spatially distributed climate vulnerability maps of the eight key crops imported by the EU. The results are all for 2050 under the RCP 6.0 concentration pathway. Soybean accounts for the EU's greatest dependency on countries outside its borders in terms of water, due to the large volumes imported. Approximately 82% of the EU's soybean imports come from Brazil, Argentina, and the United States of America (USA). These three countries also constitute the largest share in the external water dependencies of the EU related to soybean. Around 60% of soybean imports in 2050 originate from areas with a high or very high vulnerability to drought. Only 4% of soybean imports' vulnerability to drought in 2050 is lower than current climatic conditions. The climate vulnerability scores of the largest soybean exporting countries are: 1.5 for Brazil, 1.4 for Argentina, and 1.2 for the USA, indicating moderate to high climate vulnerability.

The EU is an important region for the cocoa sector and accounts for more than half of global cocoa bean imports. It is 100% dependent on cocoa imports for its chocolate industry. Most cocoa beans are supplied to the EU directly from developing countries, predominantly in West Africa. In 2017, 61% of the market share of EU cocoa imports was supplied by Ivory Coast (40%), Ghana (12%), and Nigeria (9%). Around 28% of the cocoa imports will be originating from high to very highly vulnerable locations to drought in 2050 under RCP 6.0. The vulnerability of the 19% of the cocoa imports to drought will be less in the future compared to current climatic conditions. We observed a significant escalation in drought vulnerability levels for supplies of cocoa beans from Indonesia, which has a climate vulnerability score of 3.9, and from Malaysia, which has a score of 3.0 in 2050. Supplies from the Ivory Coast and Ghana have relatively lower vulnerability scores at 1.3. In contrast, South American suppliers of cocoa will be less vulnerable to drought. Cocoa exports from Peru, Colombia, Uganda, and Gabon will benefit from climate change and be less vulnerable to drought in 2050.

The EU has a large coffee market and accounts for just under a third (30%) of global coffee consumption. Globally, the coffee export market is dominated by Brazil and Vietnam which, together, provide half of Europe's imports. Our analysis shows that coffee imports to the EU will be significantly affected by increased droughts in the future as a result of climate change. 44%

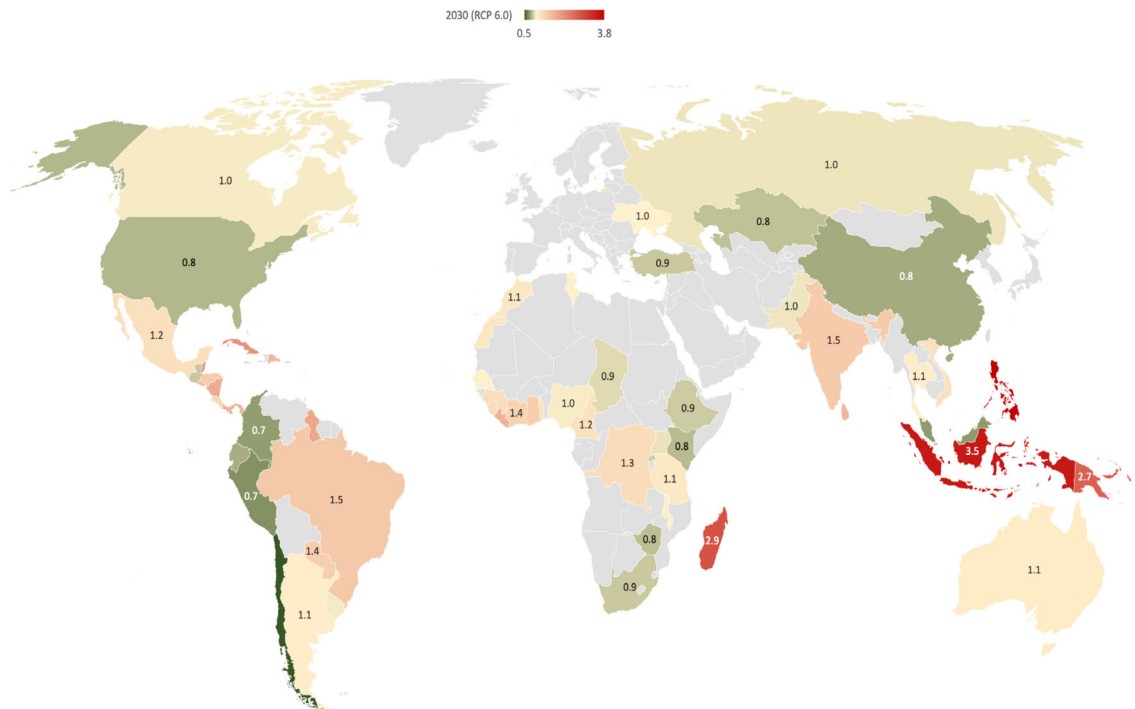

**Fig. 2 Cross-border climate vulnerability score (CCVS) of the EU's agri-food economy to drought per exporting country in 2050 under the RCP 6.0 concentration pathway.** The figure only represents countries that account for more than 0.1% of the total green virtual water import by the EU. Together, these countries represent more than 99% of the total external rainfall dependency of the EU. Green to red colors indicate CCVSs in ascending order.

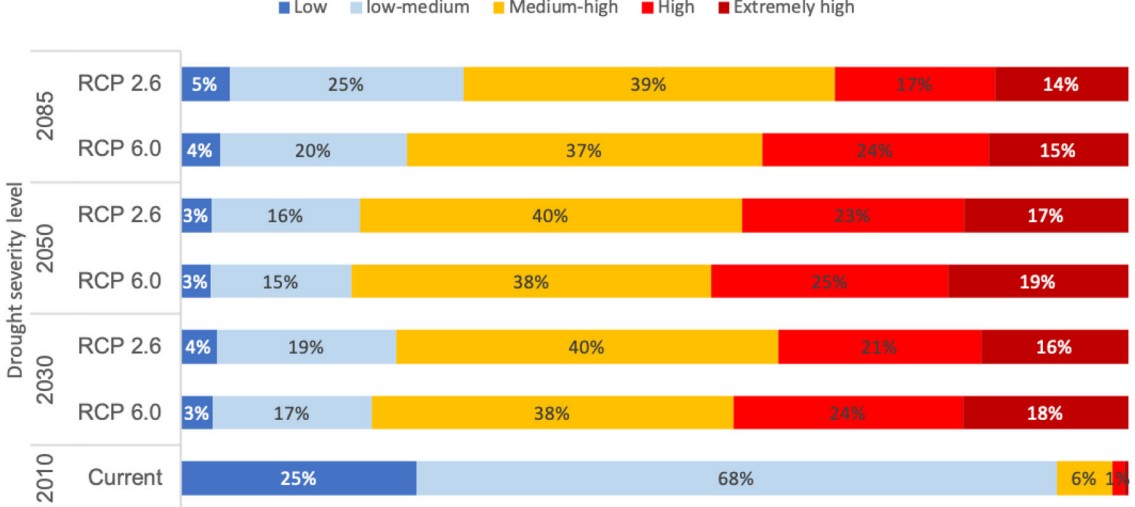

**Fig. 3 Percentage of agricultural import volume by the EU categorized by the drought severity levels at exporting locations for different climate scenarios and time slices.** Blue to red colors indicates drought severity levels.

of the coffee imports' supply locations will be highly vulnerable to drought due to climate change. Only 28% of the coffee imports' vulnerability will be less compared to the current climatic conditions. In each of the years studied, the vulnerability level of coffee imports from Europe's main suppliers, Brazil and Vietnam, are high, with climate vulnerability scores of more than 1.9. Supplies from Indonesia are also greatly affected, with vulnerability scores of around 4.5. However, imports from Colombia, Uganda, Peru, Ethiopia, and Kenya will become less vulnerable to drought under climate change.

Indonesia, Malaysia, and Thailand are major sources of EU palm oil imports. In 2017, close to three-quarters of EU imports

of palm oil came from these three Asian countries. We find that the EU's supplies from Indonesia are highly vulnerable to drought, with a climate vulnerability score of more than 3.0. Other global suppliers, Papua New Guinea and Brazil had similarly high scores. Yet EU's other large palm oil suppliers, Malaysia and Thailand, will be moderately affected by climate change, with climate vulnerability scores between 1.25 and 1.5. Overall, 61% of palm oil imports will become highly vulnerable to drought.

The EU is a significant global producer of sunflower seeds. However, its market demand exceeds its production volumes. Therefore, it imports large quantities of sunflowers seeds from

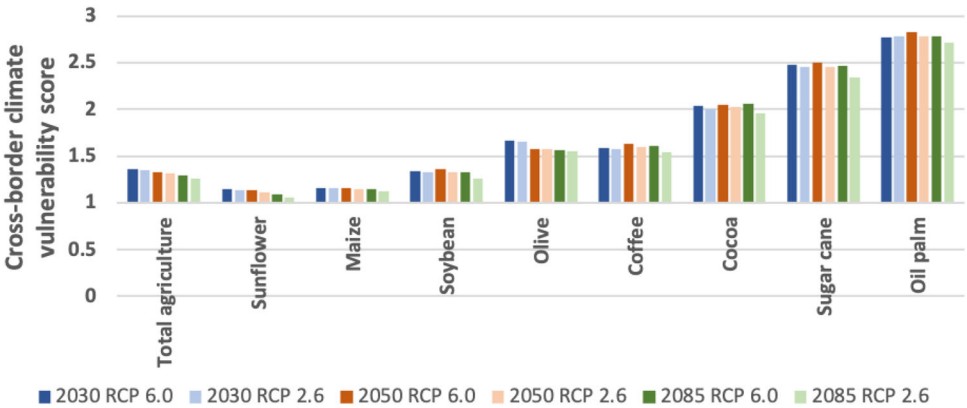

**Fig. 4 Cross-border climate vulnerability scores of the EU related to the eight most important crops imported to the EU that explain the continent's external rainfall dependency.** The colors of the bars indicate the different year and RCP combinations, 2030 RCP6.0, 2030 RCP2.6, 2050 RCP6.0, 2050 RCP2.6, 2085 RCP6.0, 2085 RCP2.6.

Ukraine, Argentina, Russia, and the USA. The external rainfall dependency of the EU related to sunflower is around 48%. Our analysis shows that almost all of the EU's suppliers of sunflower seeds have a low climate vulnerability to drought, with the exception of Bolivia and Paraguay.

The EU produces most of the maize it needs to meet demand; only 30% is imported from outside the EU and is used mainly as feed for cattle production. The major maize-exporting countries to the EU are Ukraine, Brazil, and Argentina. The climate vulnerability of all significant maize-exporting countries is low. Exports from Brazil will become less vulnerable to drought under climate change. Only around 20% of the maize imports will originate from highly drought vulnerable areas in 2050 (RCP6.0).

The EU grows most of its olives; it only imports 8% from outside its borders. Its external rainfall dependency related to olives is around 13%. Although the EU is not dependent on olive imports for its olive-based economy, its olive imports are one of the most vulnerable to climate change. The climate vulnerability of the major olive-exporting countries, such as Turkey, Tunisia, and Morocco, are highly vulnerable to drought under climate change (the climate vulnerability scores of the imports from Tunisia and Turkey are calculated as 1.5, and Morocco as 1.6).

The EU does not produce any raw cane sugar; it is all imported from other countries. Sugar cane (both in raw and processed form) comes mainly from Brazil, Mauritius, Cuba, Guyana, Fiji, and India. Our analysis shows that all sugar cane suppliers are very highly vulnerable to drought under climate change, with climate vulnerability scores higher than 2.0, with the exception of Fiji. More than 73% of the sugar cane imports will be highly vulnerable to drought by 2050 and only 6% of the imports' vulnerability will be low. This result makes the sugar cane the most climate-vulnerable imported commodity by the EU amongst other key-imported products.

**Discussion**

The main finding of this study is that the EU's economy is highly vulnerable to drought outside its borders due to climate change. The analysis shows that crop product flow through international trade means that all sectors which use raw materials through global supply chains are connected to water resources—and to extreme weather events and climate change—in the original production regions. As the intensity and magnitude of extreme weather events will alter under a changing climate, sectors in the EU that are dependent on this external product flow will become more vulnerable to extreme weather events, as demonstrated in our study for the EU's agricultural imports.

Almost all climate vulnerability studies for a geographic region focus primarily on climate impacts within the boundaries of the region. Cross-border susceptibilities have not been fully addressed in scientific literature, nor in climate-related policies and strategies. The results and analytics presented in our study could move the focus of climate vulnerability studies to a new step, including cross-border climate susceptibilities. Our study shows that the climate vulnerabilities of some sectors in the EU, such as meat and dairy, chocolate (cocoa), coffee, food, and cosmetic production based on palm oil, related to drought, mainly lie beyond the EU's borders, not within. A good example of this is palm oil. Mostly used for food, cosmetics, and biofuel production in the EU, we show that palm oil imports to the EU are highly vulnerable to drought under climate change. EU-wide climate-related strategies, such as the Climate Adaptation Strategy and the EU's agricultural trade policy, as well as international development strategies at the pan-European and regional level, can benefit from the results of our assessment. They could further address these cross-border climate vulnerabilities on a sectoral basis in order to prevent any negative consequences that the EU economy may face. The EU can also use the outcomes of our study when developing bilateral relations with trade partners.

One of the other key outcomes of our study is that the vulnerability of the EU's agricultural imports to drought sharply increase within the next twenty to thirty years (by 2030, representative of the next thirty years) for some key imported crops such as sugar cane, cocoa, coffee, and palm oil. This suggests that immediate action is needed to prevent the possible negative impacts. Adaptation is necessary at all levels of decision making, and options, such as sourcing from other regions and investing in new market areas, supporting specific regions with efforts to reduce their vulnerability thus become more drought resilient, or using alternative primary products, should be considered on a case-by-case basis. For example, a multi-national company might choose to work together with its suppliers and invest in building drought resilience in production locations outside the EU's borders, or it could consider shifting its supply chain markets to the places where vulnerabilities to drought are expected to be lower in the future. Previous studies showed the potential use of locally, inside the EU, grown feed source alternatives than imported soybeans to lessen environmental pressures and cross-border vulnerabilities[11–13]. This can be considered in further policy formulation as well. By comparing vulnerabilities under two different concentration pathways (RCP 2.6 and RCP 6.0), our study reveals a clear trend of decreasing climate vulnerabilities from RCP 6.0 to RCP 2.6 for all crops, and for most of the

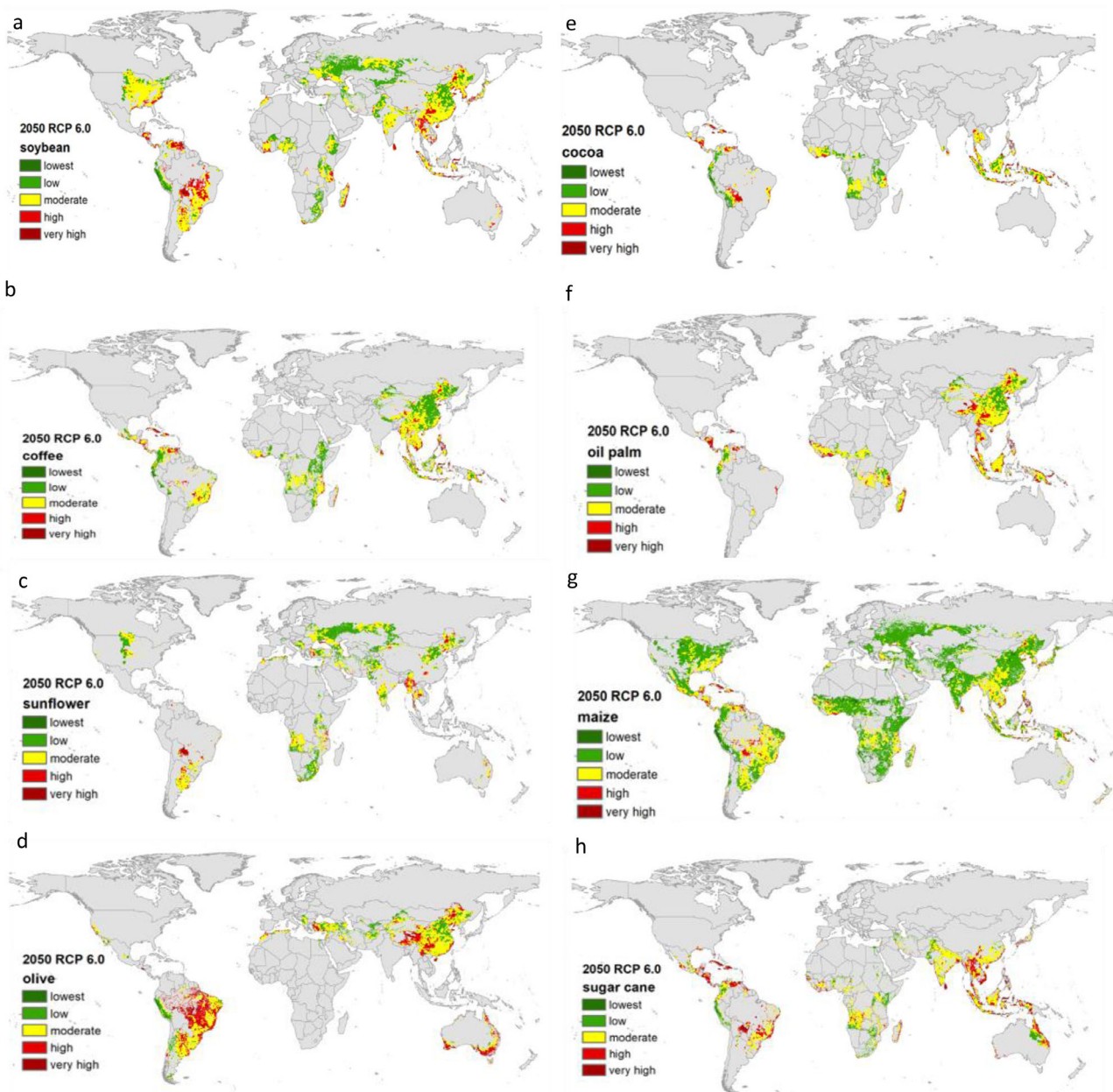

**Fig. 5 Climate vulnerability level maps of the key agricultural products imported by the EU to drought in 2050 under the RCP 6.0 concentration pathway.** Left column from top to bottom: soybean (**a**), coffee (**b**), sunflower (**c**), olive (**d**). Right column from top to bottom: cocoa (**e**), oil palm (**f**), maize (**g**), sugar cane (**h**). Green to red colors refers to vulnerability levels, from the lowest to very high.

exporting locations. This highlights the importance of global efforts to mitigate climate change and reduce greenhouse gas emissions as rapidly and dramatically as possible.

Our analysis underlines the importance of analyzing each crop and production location on a grid-scale when assessing the climate vulnerabilities of agriculture (both internal and external to the EU). This is because the vulnerability of each crop varies significantly: there is a low drought vulnerability for sunflower seeds and maize imports; a moderate climate vulnerability for soybeans and a high to very high vulnerability for coffee, cocoa, and palm oil imports. Vulnerabilities also differ per exporting location. For example, coffee imports from Indonesia, Brazil, and Vietnam are highly vulnerable to drought under climate change, whereas those from Colombia, Uganda, Peru, Ethiopia, and Kenya will be less vulnerable. Consequently, we recommend assessing climate vulnerabilities to extreme weather events per

crop, not per agricultural sector as a whole. The grid-based vulnerability maps we produced for the key imported crops show distinct differences in vulnerability levels per location in a specific country. This is particularly relevant for large-sized exporting countries such as the USA, Brazil, and China.

The determinant factor in climate vulnerability in our assessment was changes in drought conditions, in terms of both magnitude and intensity, and, thereby, in the hydro-meteorological drivers in the production locations. We expressed the effect of socio-economic drivers in terms of changes in crop import demand by the EU, which did not change significantly under the SSP2 scenario. However, the decrease in vulnerability levels in 2085, when compared to 2030 and 2050, can be explained by a decreased population in the EU, and a consequent decline in crop demand. Although socio-economic drivers such as population changes in the EU were not a determinant factor in

our study, they can be in terms of competition for food at the exporting locations. For example, the EU's imports of soybean can be affected if internal demand for the commodity increases, creating competition. This can lead to increased commodity prices and further restrictions on commodity trade. This study does not consider such additional pressures on the EU's import dependency and vulnerability.

The outcomes of the study also revealed that adaptive capacities to climate change can play a key role in reducing vulnerabilities in CCVSs as observed in the USA, Indonesia, and Malaysia. Despite higher drought severity in the future, vulnerability scores for the imports from these countries are estimated lower because of their higher climate change adaptive capacities. However, most of the agricultural imports by the EU such as soybean (Brazil) and cocoa (Ivory Coast and Ghana) are produced in areas with low adaptive capacity to climate change. Working towards more climate-resilient production in these countries can be a meaningful step for the EU in order to reduce its cross-border climate vulnerabilities.

The results presented in this study can be seen as the first steps in providing a deeper and more thorough understanding of the cross-border vulnerabilities of a region, the EU. Therefore, the analysis has some limitations that may affect the outcomes and can be further addressed in future studies. Firstly, we simplified the potential impacts of climate change only by looking at drought severity, soil moisture anomaly. There are several other factors such as length of the growing season, other stressors such as heat stress, water scarcity, frost, floods, and potential $CO_2$ fertilization. Future research on this topic could embed the multiple stressors and other factors when determining cross-border climate vulnerabilities. Another major limitation is that we have only considered existing trade patterns and the current crop water demand in the analysis. Consequently, our analysis does not consider production shifts that may happen between regions due to sudden production losses or price alterations in the future. By keeping the water demand of crops constant, we have not considered the climate impacts on crop water use and evaporative demand. This means that the climate impacts are simplified. The crops' vulnerability to drought conditions may alter, because climate change may result in higher, or lower, water demand for crop production.

Models have been increasingly used to explore development pathways in food systems under a range of scenarios and in a holistic way[38]. Our work focused on the trade dimension and drought vulnerabilities under climate change. Nexus and system approaches are considered to be essential for progress toward meeting the Sustainable Development Goals (SDGs)[39]. For this, our approach and outcomes can be complemented with other climate-related hazards (floods, pests, diseases, etc.) while assessing the vulnerability of value chains of agricultural products. This system thinking can be key for finding sustainable solutions pertaining to climate adaptation and cross-border vulnerabilities[40].

Climate change is leading to increased drought in many parts of the world, such as in Southeast Asia, Sub-Saharan Africa, and South America. This has implications for the EU's economy because a lot of the goods it consumes or uses are produced in these regions, and they will be at risk because of climate-induced drought severity. This makes the EU's economy highly vulnerable to drought under climate change well beyond its borders. Our work mapped the dependencies of the EU's economy on water resources outside its borders and quantified its cross-border climate vulnerabilities. We found that more than 44% of the EU's agricultural imports will become highly vulnerable to drought in the future because of climate change. The drought severity in production locations of the agricultural imports will increase around by 35% in 2050 (RCP 6.0 case). This is particularly valid for imports that originate from Brazil, Indonesia, Vietnam, Thailand, India, Turkey, and Honduras. Although climate change will negatively impact these locations, some exporting locations will benefit from changes in rainfall patterns. For example, imports from Russia, Nigeria, Peru, Ecuador, Uganda, and Kenya will be less vulnerable to drought under climate change.

Our study also concludes that, in the near future, supplies of certain crops to the EU could be disrupted due to increased drought in other parts of the world. Coffee, cocoa, sugar cane, oil palm, and soybean are the most climate-vulnerable imported agricultural products by the EU. A large portion of these imports will come from areas with high drought severity in the future. The EU's economic dependency on goods produced in regions that are vulnerable to water-related climate impacts can be considered in government policies and business strategies. The strategic importance of some regions, such as Southeast Asia and South America, will increase for the EU regarding the potential climate-induced impacts on water resources and the need for a continuous supply of commodities from these regions. Investments, such as increasing drought resilience and strengthening water governance to ensure sustainable, efficient, and equitable water use, could reduce the cross-border climate vulnerability of the EU's economy. The EU's policy and producers can also find alternative production options, e.g. alternative locally grown feed sources other than soybeans, to reduce their cross-border climate vulnerabilities.

## Methods

The approach to vulnerability used in this study stems from an assertion by Turner et al.[41] that vulnerability is determined by the degree to which a system will experience stress due to a given pressure, or to a combination of pressures. We expressed the cross-border climate vulnerability of the EU's economy ($V_{EU}$) as a function of change in sensitivity ($\triangle S$), change in exposure ($\triangle E$) to hydrological

**Table 1 Drivers and assumptions used in virtual water trade scenario model in this study.**

| Driver | Elements | Future conditions |
|---|---|---|
| Population growth[36] | Population size | Medium fertility (SSP2) |
| Economic growth[36] | Income levels | Medium, the current trend |
| | GDP growth | SSP2 |
| Consumer preferences[42] | Diets | Current trend |
| | Fiber demand[48] | Current trend |
| | Non-food demand[48] | Current trend |
| Production and trade[42] | Production and import ratios | Based on A1B production and T1 trade patterns |
| Technology development[42] | Water productivity | Current |
| Policy change[43] | Trade policy | Weak globalization |
| | Environmental policy | Both reactive and proactive |
| | Biofuel policy | Current |

extremes under different climatic conditions, and adaptation capacity ($A$):

$$V_{EU} = \triangle S \times \triangle E \times A \tag{1}$$

To quantify the cross-border climate vulnerabilities of the EU's economy, we first estimated the change in sensitivity to drought under climate change (i.e. change in green virtual water imports by the EU) for the chosen years of analysis: 2030; 2050; and 2085. We did so by constructing a global demand-production scenario based on a number of drivers of change: population growth; economic growth; production/trade patterns; and consumption patterns (e.g. dietary preferences, bioenergy use, etc.), based on a virtual water trade scenario model developed by Ercin and Hoekstra[42,43] (Table 1). For this, we used the shared Socioeconomic Pathway 2 (SSP2) characteristics, with the assumption that the green water footprint of crops in the locations where they are grown would remain unchanged.

The next step was to identify which imported crops to use as a basis for the vulnerability assessment in addition to the total agricultural imports. Crops with green virtual water import volumes larger than 2% of the EU's total green virtual water import were identified as the key imported products for the target years of this study. Having identified and mapped the production locations of the key imported crops on a $0.5 \times 0.5$-degree grid-scale, we estimated their exposure to drought severity in the production locations under two Representative greenhouse gas Concentration Pathways (RCP): RCP 2.6 and RCP 6.0. We did this by using an ensemble of four General Circulation Models (GCM) and four Global Hydrological Models (GHM) (4 GCMs × 4 GHMs). We converted the model outputs into groups of statistics (mean, median, upper/lower-bound, such as Q25 and Q75) and used the median values to present our results.

RCP 2.6 has been described in the literature as the best case for limiting anthropogenic climate change. It represents a scenario in which global warming is limited to below 2 degrees Celsius. RCP 2.6 was developed by the IMAGE modeling team of the PBL Netherlands Environmental Assessment Agency. It is a peak-and-decline scenario; its radiative forcing level reaches a value of around 3.1 W/m$^2$ by mid-century and returns to 2.6 W/m$^2$ by 2100. In order to reach such radiative forcing levels, GHG emissions, and indirect emissions of air pollutants, are reduced substantially over time[44,45].

In contrast, RCP 6.0 represents a scenario in which GHG concentrations double by 2060 and then dramatically fall but remain well above current levels. RCP 6.0 was developed by the AIM modeling team at the National Institute for Environmental Studies (NIES) in Japan. Projections for temperature according to RCP 6.0 include continuous global warming in which temperatures rise by about 3–4 °C until 2100. It is a stabilization scenario; total radiative forcing is stabilized shortly after 2100, without overshoot, by the application of a range of technologies and strategies for reducing GHG emissions[46,47].

**Sensitivity: green virtual water imports.** In this study, we defined sensitivity to hydrological extremes related to an imported product $p$ for year $y$ ($S_{p,y}$) as equal to the green virtual water import (VWI$_{eu,green,p,y}$ in m$^3$/year) by the EU in year $y$:

$$S_{p,y} = VWI_{eu,green,p,y} \tag{2}$$

The green virtual water import by the EU (VWI$_{eu, green, p, y}$), in m$^3$/year, for the product $p$, is the sum of the green virtual water import related to product $p$ by all EU Member States from all the countries outside the EU (non.eu) in year $y$:

$$VWI_{eu,green,p,y} = \sum_{e=1}^{non.EU} \left( T_{EU,p,e,y} \times WF_{green,p,e,y} \right) \tag{3}$$

$T_{EU,p,e,y}$ is the physical quantity of the imported product $p$ (tonne/year) by the EU from exporting country $e$ in year $y$, and WF$_{green,p,e,y}$ is the green water footprint (m$^3$/tonne) of the imported product $p$ in the exporting country $e$ in year $y$. The green water footprint volumes are taken from Ercin et al.[8].

The physical quantity of the imported product $p$, $T_{EU,p,e,y}$ (tonne/year), from exporting country $e$ to the EU is calculated as:

$$T_{EU,p,e,y} = T_{EU,p,y} \times f_{e,p} \tag{4}$$

where $T_{EU,p,y}$ is the total volume of imports related to product $p$ for the year $y$ and $f_{e,p}$ is the share ratio of the country $e$, which was calculated as follows:

$$f_{e,p} = \frac{e_p}{E_p} \tag{5}$$

$$E_p = \sum_{e=1}^{non-EU} e_p \tag{6}$$

$e_p$ is the import volume of product $p$ from country $e$ and $E_p$ is the total import volume for the same product, calculated for the year 2010 (average for 2005–2013).

The total volume of imports of the product $p$ is calculated as the difference between total *demand* ($D_{EU,p,y}$, *in tonne/year*) for the product $p$ by the EU and production of the product $p$ in the EU ($PR_{EU,p,y}$ *in tonne/year*) for year $y$:

$$T_{EU,p,y} = D_{EU,p,y} - PR_{EU,p,y} \tag{7}$$

**The demand for agricultural products by the EU.** The demand of an agricultural product $p$ by the EU, $D_{EU,p,y}$ in year $y$ has three components: (i) demand for

consumption for food $P_{f,EU,p,y}$; (ii) demand for non-food products (fiber, feed, and biofuel), $P_{nf,EU,p,y}$; and (iii) demand for an agri-food export of the EU, $P_{E,EU,p,y}$:

$$D_{EU,p,y} = P_{f,EU,p,y} + P_{nf,EU,p,y} + P_{E,EU,p,y} \tag{8}$$

The food demand $P_{f,EU,p,y}$, in tonne/year, by the EU related to product $p$ for the year $y$ is defined as:

$$P_{f,EU,p,y} = pop(EU, y) \times kcal(EU, p, y) \times f_{\frac{kg}{kcal},p} \tag{9}$$

where pop$(EU, y)$ is the population of the EU in year $y$ and kcal$(EU, p, y)$ is per capita kilocalorie intake related to product $p$ in the EU in year $y$. The coefficient $f_{\frac{kg}{kcal},p}$ is the amount of kilocalories per kilogram of product $p$. Kilocalorie values per unit product mass for the product $p$ in year $y$ are obtained from Ercin and Hoekstra[42,43]. We used SSP2 population projections from the International Institute for Applied Systems Analysis (IIASA)[36].

The non-food consumption of the agricultural product $p$, $P_{nf,EU,p,y}$, tonne/year, in the EU for year $y$ is defined as:

$$P_{nf,EU,p,y} = pop(EU, y) \times f_c(EU, p)\big|_{y=2010} \tag{10}$$

where $f_c(EU, p)\big|_{y=2010}$ is the per capita demand for the product $p$ in the EU for non-food purposes in 2010 and obtained from FAO[48].

The agri-food demand for the export industry by the EU is assumed to be proportional to the sum of food and non-food demand, defined as:

$$P_{E,EU,p,y} = \frac{P_{E,EU,p,y=2010}}{\left( P_{f,EU,p,y=2010} + P_{nf,EU,p,y=2010} \right)} \times \left( P_{f,EU,p,y} + P_{nf,EU,p,y} \right) \tag{11}$$

where $P_{E,EU,p,y=2010}$ is the amount of export of the product $p$ by the EU in 2010 (average of 2005–2013), obtained from the International Trade Centre database[49,50].

**Production of agricultural products in the EU.** The expected production of product $p$ (tonne/year) in the EU, $PR_{EU,p,y}$, is calculated as a multiplication of the commodity production share $f_{p,EU,y}$ and the total production of the product $p$ in the world, $PR_{p,y}$:

$$PR_{EU,p,y} = PR_{p,y} \times f_{p,EU,y} \tag{12}$$

Commodity production shares of the EU related to product $p$ for year $y$ and are taken from Ercin and Hoekstra[43]. The global production volumes for the product $p$ (PR$_{p,y}$) is taken from the same study, then adjusted with the global population forecasts of SSP2.

**Exposure: drought severity under climate change.** To quantify exposure, we estimated the spatial distribution of drought severity at the $0.5 \times 0.5$-degree grid-scale following the methodology by Sheffield and Wood[51]. The authors defined drought occurrence as an extended period of anomalously low soil moisture and as a consecutive sequence of months of length $D$ with soil moisture quantile values,, $q(\theta)$, less than a chosen threshold, $q_0(\theta)$. Here we chose the threshold value of 20%, which reflects conditions that occur only once every five years for a particular month, on average. Drought severity, SE, is then calculated based on duration $D$, the intensity $I$, and severity SE that are dependent on $q_0(\theta)$:

$$SE = DxI \tag{13}$$

$$I = \frac{1}{D} \left( \sum_{t=t1}^{t+D-1} q_0(\theta) - q_t(\theta) \right) \tag{14}$$

Intensity is the mean magnitude over the duration of the drought, and severity is the time-integrated deficit below the threshold, with units of %months.

We estimated drought severity at a $0.5 \times 0.5$ degree spatial resolution from 2006 to 2099, using the monthly soil moisture outputs for four GHMs: H08[52], LPJmL[53] PCR-GLOBWB[54,55], and WaterGAP2[56]. For each model or ensemble member, we calculated the severity of drought in 30-year periods centered at 2030, 2050, and 2085 s.

To assess climate impacts, we used outputs of four GCMs: GFDL-ESM2M; HadGEM2-es; IPSL-CM5A-LR; and MIROC5, being used as forcing to the GHMs. These GCMs were bias-corrected following Hempel et al.[57] and Lange et al.[58]. They represented two GHG concentration pathways (low: RCP 2.6, high: RCP 6.0) and one pre-industrial controlled run as set-up under the ISIMIP2b[59] framework. Ensemble results were used as an input for ensemble statistics. We used the drought severity values that represent the median values from the ensemble.

**Adaptive capacity to climate change.** To express adaptive capacity to climate change for each exporting country, we used datasets provided University of Notre Dame Global Adaptation Index (ND-Gain Index)[60]. The dataset provides adaptive capacity for the agriculture sector based on four indicators: capacity to equip agriculture areas with irrigation, N+P205 total fertilizer use on arable and permanent crop area use, pesticide use, and tractor use. Adaptive capacity scores per defined per country between 0 and 1 showing baseline minimum (lower scores

**Table 2 Climate vulnerability scores and corresponding vulnerability levels.**

| Climate vulnerability score (CCVS) | Climate vulnerability level |
|---|---|
| CCVS ≤ 0.5 | Lowest |
| 0.5 < CCVS ≤ 1 | Low |
| 1 < CCVS ≤ 1.5 | Moderate |
| 1.5 < CCVS ≤ 2 | High |
| CCVS > 2 | Very high |

reflect higher adaptive capacity) and maximum. We have used the most recent year of the ND-Gain Index, referring to 2018.

**Vulnerability assessment**. We calculated cross-border climate vulnerability of the EU's economy, ($V_{EU,p,e,y}$), related to an imported product $p$ from an exporting location $e$ in year $y$, as a function of change in exposure ($\Delta E_{p,e,y}$), change in sensitivity to hydrological extremes ($\Delta S_{p,e,y}$) and adaptive capacity of the exporting location (Ae):

$$V_{EU,p,e,y} = \sum_{i=1}^{n} \left( \Delta E_{p,e,y,i} \times \Delta S_{p,e,y,i} \times \frac{S_{p,y,i}}{S_{p,y,e}} \times Ae \right) \quad (15)$$

$i$ represents a grid cell located in an exporting location $e$, which has $n$ number of grid cells. $\frac{S_{p,y,i}}{S_{p,y,e}}$ refers to the ratio between grid's sensitivity to exporting country's sensitivity. This ratio was introduced to distribute demand changes from exporting country to grid cell in each county. The demand change is calculated at a country level and drought severity is at the grid level. To calculate vulnerability level at grid-scale, this ratio was used as a distribution factor.

Change in exposure related to a product $p$ imported by the EU from an exporting location $e$ for the target year $y$ ($\Delta E_{p,e,y}$) is calculated as the ratio of drought severity at the production location of the imported product $p$ between the target year $y$, $E_{p,e,y}$ in %month, and 2010, $E_{p,e,y=2010}$ in %month:

$$\Delta E_{p,e,y} = \frac{E_{p,e,y}}{E_{p,e,y=2010}} \quad (16)$$

Change in sensitivity to hydrological extremes related to a product $p$ imported by the EU from an exporting location $e$ for the target year $y$ ($\Delta S_{p,e,y}$) is calculated as the ratio of green virtual water import by the EU related to product $p$ from between the target year $y$, $VWI_{e,green,p,y}$ in m³, and 2010, $VWI_{e,green,p,y=2010}$ in m³:

$$\Delta S_{p,e,y} = \frac{VWI_{e,green,p,y}}{VWI_{e,green,p,y=2010}} \quad (17)$$

Cross-border climate vulnerability demonstrates how much green virtual water import volumes (in m³) and drought severity together change in an exporting location (country, region, or globally) under different climatic conditions compared to current climate characteristics. A climate vulnerability score of 1 for an exporting location (e.g. a grid cell, region, country, or global) means that future vulnerability to hydrological extremes (in this study to drought severity) in the target year is the same as of today (2010). Scores below 1 indicate a decreased vulnerability to drought and above 1 show an increased vulnerability to drought compared to current climatic conditions.

To map the level of vulnerability to drought for the target years, we also set thresholds for vulnerability, CCVS ≤ 0.5 (lowest level) and CCVS > 2 (highest level) (Table 2). According to the vulnerability numbers, we defined five levels of vulnerability: lowest, low, moderate, high, and very high. The lowest category means vulnerability to drought is significantly reduced in the future because of reduced green virtual water import and/or decreased drought severity. Low vulnerability level indicates climate vulnerability is close to the current levels and climate impacts are not significant. Moderate vulnerability level means climate change negatively affects drought severity in the production locations. High and very high vulnerability shows significantly increased drought severity in production locations, as well as larger green water virtual water import volumes.

**Reporting summary**. Further information on research design is available in the Nature Research Reporting Summary linked to this article.

## Data availability
The datasets generated during and/or analyzed during this study are available from the corresponding author on reasonable request.

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

## Acknowledgements

This research study was funded by the European Union's Horizon 2020 research and innovation program under the IMPREX and RECEIPT projects (grant no. 641811 and grant no.820712).

## Author contributions

E.E. and J.H. designed the research. T.V. collected soil moisture and climate change data and modeled the drought severity. T.V. performed the statistical analysis. E.E. performed the vulnerability analysis and virtual water trade calculations. E.E., T.V., and J.H. wrote the paper. All authors approved the final version.

## Competing interests

The authors declare no competing interests.
