## [Peer Review File · Nature Communications]

REVIEWER COMMENTS

Reviewer #1 (Remarks to the Author):

This study raises an important point of the dependency of European agricultural imports from non-EU countries on drought severity on crops. It shows that in future the climate vulnerability of the EU's food security enhances. This will increase the pressure on the EU's agro-food economy and clearly request to limit climate change. The only easing comes from the negative population growth in Europe, but this will not be the case in countries from which agricultural imports come. That gives additional pressure on the agricultural sector which might have to be taken into account in this study as well.

The paper is part of a series of papers by the authors in the investigation of the water footprint, whereby the present study clearly focuses on the endangerment of European agricultural imports due to droughts. The novelty of the methods are small, but the key aspect of drought severity of cross-border commodities is very useful. The paper is well written and in most parts generally understandable.

Comments on methods:

Authors use DGVMs/GHMs to estimate drought severity by using soil moisture calculations, but it would easily be possible to estimate drought stress of crops directly as well as the water footprint provided by the models. Furthermore, this would be more consistent as the water footprint is not influenced by future climate and CO2 changes in the present study. Is there a reason why you do not use these data?

I'm wondering how diet changes develop over the investigation period. This information would help to understand results better. Soybeans are often mentioned in the paper, which are mainly consumed by the animal stock. As this will be reduced in future, it would be interesting to know how this influences future imports. Some supplementary material about all the assumptions would be very helpful.

Minor comments:

lines 380-384 with this method and only 4 GHMs isn't it the case that the 2 most extreme models are not being taken into account for the median or does it differ between regions? Have you checked that?

lines 426-428 was very hard to understand, please revise

figure 5 legend in maps are not readable

line 162 typo: chocolate

Reviewer #2 (Remarks to the Author):

Review of MS Cross-border climate vulnerabilities of the European Union to drought by Ercin et al.

The manuscript presents a welcome approach to better understand the cross-boarder climate vulnerabilities at sectoral level. The study demonstrates that as a result of the dependency on imported agricultural primary products, the EU economy is indirectly exposed to climate risks in the production regions outside EU. This is done with means of introducing and applying a novel quantitative assessment method and mapping of crop- and geographic region -specific vulnerability index with respect to different time-scales and emission scenarios. The quantification of vulnerability as well as assessments of agricultural sector vulnerability with the level of specifying the impact on the key-crops is often lacking in studies that focus on assessing climate risks as well as adaptation needs and options.

The paper fits the scope of the journal as it presents a truly multidisciplinary study that represents a novel high-quality integrated assessments of environmental and economic scenarios that are of interest to scholar from the fields of Earth sciences. The presented findings on the vulnerability of

However, the discussion section very scarcely presents references of such research that the findings contribute to (see detailed comments below).

Also, since authors only here clarify that these limitations may effect the outcomes of the study, this point should be presented earlier with short notion (while highlighting that these results are a first step in this important next level of understanding the broader-crossing vulnerabilities).

The crops assessed in the study are based on that they are currently imported in volume and they are dependent on the EU's external rainfall dependency (row 136). Some of these are such that may be replaced by other products (e.g. soybean with broad bean) (see e.g. Sandsröm et al 2018). This is addressed briefly in discussion (row 266). To relate the discussion more strongly to relevant on-going scientific discussions literature on transformative adaptation might be found interesting.

Conclusions

row 330: authors state ' We found that more than 44% of the EU's agricultural imports will become highly vulnerable to drought in future because of climate change.' Similar to the percentage (80%) presented in the abstract, it is unclear how this number is drawn (is it an average of the proportion of all six scenarios assessed and for the crops assessed in this study, or another method, or from another source?).

Other minor comments

The paper would benefit from language check by an expert. Some challenging formulations of sentences make it hard to grasp the meaning of the text at points, for example sentences in rows 29-31; 43-45; 193-194. In sentence in rows 29-31, for example, it is not clear which studies are focused on the 'consumer perspective' and how does this relate to the cross-boarder aspect? While I like how section 'Cross-border climate vulnerability of the EU: key-imported crops' is written, the presentation of results of the different crops appear somewhat inconstant leaving an unfinished impression. E.g. the term presented in row 161: 'key player in the cocoa sector' would benefit from specifying what is meant by this, particularly for the less economy-wise reader. The following paragraph on coffee-related results is more easy to grasp. Similar to how societal change (in values/preferences that affect the demand) is speculated to impact on sugar cane (rows 218 – 220), there is literature on similar processes related to e.g. how animal product demand might change and thus reflect on maize and soybean.

row 262: I suggest not use term mitigate in relation to climate risks. It is easily confused with climate change mitigation and thus doesn't benefit the papers clarity to wider audience.

Methods

Research methods seem to be appropriate and explained clearly enough in the Method and Data section. As the quantitative methods used in this paper are not my field of expertise, I would like to present a comment from the perspective a less experienced quanti-method reader:

The temporally far-reaching results that the study presents are much needed in picturing the potential futures that such large economies as EU and it's vital sectors such as agriculture need to be prepared for. These results, however, inevitably include high level of uncertainty – a reason why earlier studies have claimed such assessments impossible referring to e.g. the uncertainty in crop water requirement changes due to productivity increase (see e.g. Sandström et al. 2018). It remains unclear how these uncertainties are addressed in this paper for an unexperienced reader of the methodology.

Reporting summary file seems to be sufficient and correct.

REFERENCES

- Grundstroem, F. and Juhola, S., 2019. A framework for identifying cross-border impacts of climate change on the energy sector. *Environment Systems and Decisions*, 39(1), pp.3-15.
- Sandström, V., Lehtikoinen, E. and Peltonen-Sainio, P., 2018. Replacing imports of crop based commodities by domestic production in Finland: potential to reduce virtual water imports. *Frontiers in sustainable food systems*, 2, p.67.

Reviewer #3 (Remarks to the Author):

The paper presented climate change vulnerability of different EU imports based on exposure and sensitivity of drought in regions of production. The methodology of using green virtual water imports as a base to determine vulnerability of EU crop imports is both a novel and interesting attempt. Most of the earlier studies focused on direct impact of climate change on crop yields and

production. The paper concluded that some of the EU crop imports such as cocoa, coffee and sugarcane will be highly vulnerable to drought in the exporting country and suggested that it will have implications on EU agricultural economy. This paper is relevant and can be very useful to academics, researchers and policy makers working on agriculture economy, international trade and food security. It has potential to generate good discussions on these topics. The use of green virtual water volume to examine the impacts of climate change on EU crop imports can also be used to measure sustainable agricultural imports to the EU. There, however, are few issues in the methodology and analysis that I believe needs to be addressed to improve the paper for publication.

The main issue I am concerned with is the way impact of climate change crops is addressed in this paper. The authors based their analysis in examining vulnerability of crop imports over drought events in exporting regions considering only the impact of drought on quantities of crop produced (represented by virtual green water volume) and exported. Crop production under climate change are not affected by soil water contents (in case of drought) alone but also by several other factors such as length of growing season, temperature and potential CO2 fertilisation etc. There are many studies which showed that crop biomass accumulation (including water content) is highly sensitive to some of these factors (Tubiello et al, 2000; Donatelli et al 2015; You et al, 2009; Challinor et al, 2014). The effects of these factors on crop production are mostly manifested together, synergistically sometimes, so the impacts of climate change would be misleading if effect of only one factor is isolated and used in an analysis. Another, somewhat similar, issue is on adaptations under climate change. Farmers are known to make small adjustments to mitigate any adverse production conditions. It is argued by many researchers that a climate impact study without a provision of potential adaptation will be highly exaggerated (Medelsohn et al, 1996; Tan and Shibasaki, 2003; Wreford et al, 2010 etc). Will the results in this paper still show a higher vulnerability to some crop imports, for instance, if the green virtual water volume is decreased but there is an increased use of irrigated water to minimise loss in production? IPCC definition of vulnerability, as stated by the authors on line 27-29, includes adaptive capacity as one of the essential functions to measure vulnerability. That is why many climate change studies included potential adaptations in the analysis of climate change impacts on agriculture (Wreford and Adgar, 2015; Donatelli et al, 2015 etc.). The authors acknowledged that climate change factors other than drought effects and adaptations to minimise those effects are not included in the paper. Nevertheless, in their conclusions, the authors suggested the results and analysis presented in this paper would be useful to the EU policy makers developing climate change strategies and trade bilateral relationships. I believe to make the results and analysis presented in the paper stronger and more justifiable, the authors need to adjust the climate change vulnerable scores to account overall climate change impacts and potential adaptations on crop production the exporting regions. The main theme of the paper is exploring the impacts of future drought events under climate change. However, there is not much review on past drought incidents and their impacts on the EU agriculture much except for a media report at the start of the introduction. There are plenty of reports and studies that shows the severity, intensity and impacts of droughts in different years in the EU (Spinoni et al., 2016; Ciscar et al 2014; Wreford and Adger, 2011; Benzie et al., 2011; Garcia-Herrera et al 2010 etc.) and elsewhere (Qureshi et al 2013; Medellin-Azura et al., 2016). A thorough review of some of these studies will provide a good insight to the objectives of the paper. The statement on line 29-30 indicates that the current paper is looking at climate vulnerability through consumer's perspective. However, I feel that the analysis presented in this paper is still production based as it uses CCVSs that only includes virtual water volume (in other words quantity of production). A consumer's perspective will include availability (supply), accessibility (potential disruption in supply chain due to climate change effects such as flooding) and affordability (increase in price due to shortages) which was not what the paper was aiming for. Figure 1 presents CCVSs disaggregated over two drivers; socio-economic and meteorological (drought). I thought CCVSs were determined based on drought vulnerabilities (which is the main theme of the paper). I wonder how CCVSs are disaggregated when the base is one of the drivers. Keeping that aside, the authors also suggest a decrease in CCVSs in 2085 compared to 2030/2050 is due to decrease in population growth in the EU over that period. This is understandable in Figure 1 but how the CCVSs also decrease in 2085 for drought vulnerability driver. Do the future drought events also decrease over time?

Besides these issues, there are some minor specific issues which are listed as follows;

Abstract: First line – Shouldn't it be 'European Union' rather than Europe Union.

Line 7 – Just to mention that crop import vulnerability is not only concern agricultural economy but

Cross-border climate vulnerabilities of the European Union to drought

Response to the reviewers

Ertug Ercin ^{a,b}, Ted I. E. Veldkamp^c, Johannes Hunink^d

a R2Water Research and Consultancy, Amsterdam, the Netherlands

b Institute for Environmental Studies, Vrije Universiteit Amsterdam, the Netherlands

c Amsterdam University of Applied Science, Amsterdam, the Netherlands

d FutureWater, Cartagena, Spain

*Correspondence: ercin@r2water.nl; Tel.: +31617366 472

Reviewer #1

This study raises an important point of the dependency of European agricultural imports from non-EU countries on drought severity on crops. It shows that in future the climate vulnerability of the EU's food security enhances. This will increase the pressure on the EU's agro-food economy and clearly request to limit climate change. The only easing comes from the negative population growth in Europe, but this will not be the case in countries from which agricultural imports come. That gives additional pressure on the agricultural sector which might have to be taken into account in this study as well.

The paper is part of a series of papers by the authors in the investigation of the water footprint, whereby the present study clearly focuses on the endangerment of European agricultural imports due to droughts. The novelty of the methods are small, but the key aspect of drought severity of cross-border commodities is very useful. The paper is well written and in most parts generally understandable.

We would like to thank the review for his/her constructive comments and positive consideration of our paper. The food demand by third countries, countries outside the EY, in the future was addressed in our demand/production scenario for 2050, since it was constructed globally. Thus, food demand was implicitly addressed in our calculations. However, as indicated by the reviewer, we did not include potential pressure created by global demand for crops (for the ones analyzed). This topic was discussed in the discussion section, please see lines 317-320 in the revised manuscript.

Comments on methods:

Authors use DGVMs/GHMs to estimate drought severity by using soil moisture calculations, but it would easily be possible to estimate drought stress of crops directly as well as the water footprint provided by the models. Furthermore, this would be more consistent as the water footprint is not influenced by future climate and CO₂ changes in the present study. Is there a reason why you do not use these data?

We understand the reason of this question raised by the reviewer. Our analysis assessed the drought severity, in terms of soil moisture anomaly, at production locations. We used this indicator as a proxy to "drought stress" experienced by crops, per crop type. We have three major reasons for using the soil moisture anomaly rather than estimating crop drought stress (thus potential yield changes per crop type):

- 1) First, we neither developed nor run any model to calculate soil moisture at grid-cell level. We calculated soil moisture anomalies from existing model runs of the ensemble of models participating in the ISIMIP project. Our goal was to use outputs of several models (in combination with various GCMs under different scenarios), so that we could obtain a relatively reliable drought severity indicator for the analysis.
- 2) The ISIMIP participating models that we used incorporated only yield losses for four crop types, namely: soybean, rice, wheat and maize. The drought stress related data for the rest of the crops addressed in our paper were not included in the model runs, so were not available. Current global models can model such drought stress response approximately for 6-10 crop types. So, of the key-imported commodities we analyzed, it would have been possible to calculate drought stress only for some of the annual crops (e.g., soybean, maize, wheat, rice etc.) and none of the perennial crops such as cocoa, coffee, olives. As far as we know, no global model runs are available today that address all these crops. This limitation, not having crop specific global models for all crop types, made us decide to use a proxy indicator (i.e., soil moisture anomaly) that we can apply to all type of crops.
- 3) To re-calculate the water footprints of crops under climate change, we had to know yield changes for each crop per climate change scenario. Following the logic of our argument described above, this was not doable considering modeling requirements and existence of such models.

Since it is not possible to model drought stress for each crop addressed in our study and unavailability (and non-existence) of such global crop models for crops like cocoa, coffee, olive in the ISIMIP database, we chose to use the indicator of soil moisture anomaly (in terms of drought severity) as a proxy for drought stress of crops. Consequently, our analysis is not an impact study but a vulnerability (proxy to actual losses) type.

I'm wondering how diet changes develop over the investigation period. This information would help to understand results better. Soybeans are often mentioned in the paper, which are mainly consumed by the animal stock. As this will be reduced in future, it would be interesting to know how this influences future imports. Some supplementary material about all the assumptions would be very helpful.

In the method section Table 1, we provided assumptions regarding dietary preferences. We chose to have the current trend, which means keeping per capita dietary choices (in terms of kcal and kgs of food item consumed) constant for future (we used SSP2 scenario). Thus, change in dietary preferences did not play a role in our analysis. Population based consumption changes, export economy (e.g., dairy and meat exports by the EU), biofuel, raw material needs were the reasons behind import/export commodity volumes.

To provide further information, we added our estimates for annual growth rate of imports by the EU per crop group in the Supplementary Information section (Table S.2)

Minor comments:

lines 380-384 with this method and only 4 GHMs isn't it the case that the 2 most extreme models are not being taken into account for the median or does it differ between regions? Have you checked that?

We used outputs from all the models. These lines refer to statistical operation of the outputs. Although we calculated, we chose not to include extremes (e.g., Q75) in our analysis but the median one (median statistics of all output numbers). These median statistics are location specific, hence all GHMs have been applied in the analysis. From climate change scenarios perspective, we did not use RC 4.5 and RC 8.5 and went for middle severity scenario of RCP 6.0 and 2.6 as the optimistic.

lines 426-428 was very hard to understand, please revise

revised.

figure 5 legend in maps are not readable

legend size was increased.

line 162 typo: chocolate

corrected.

Reviewer #2 (Remarks to the Author):

Review of MS Cross-border climate vulnerabilities of the European Union to drought by Ercin et al.

The manuscript presents a welcome approach to better understand the cross-boarder climate vulnerabilities at sectoral level. The study demonstrates that as a result of the dependency on imported agricultural primary products, the EU economy is indirectly exposed to climate risks in the production regions outside EU. This is done with means of introducing and applying a novel quantitative assessment method and mapping of crop- and geographic region -specific vulnerability index with respect to different time-scales and emission scenarios. The quantification of vulnerability as well as assessments of agricultural sector vulnerability with the level of specifying the impact on the key-crops is often lacking in studies that focus on assessing climate risks as well as adaptation needs and options.

The paper fits the scope of the journal as it presents a truly multidisciplinary study that represents a novel high-quality integrated assessments of environmental and economic scenarios that are of interest to scholar from the fields of Earth sciences. The presented findings on the vulnerability of EU agriculture-based economy are also of interest to scholars with foci on climate change adaptation and agri-economics; as well as to EU sectoral and climate policy makers, and, to an extent, to agricultural practitioners. The study addresses a theme of widespread interest i.e. how climate risks may realize in future with significant consequences. The study presents this in a way that makes it understandable also to a wider public.

The research is presented with convincing arguments. Conceptually the manuscript focuses on cross-broader climate change vulnerability, drought severity, and virtual water (trade/dependency). These concepts are well operationalised and are used with sufficient consistency to reach conclusions.

We would like to thank the reviewer for his/her very positive and constructive comments.

The main handicap of the work is that it scarcely presents references of on-going debates on the field that the findings contribute to. To reach the full potential of this important article, minor, while numerous, issues in the paper need to be revised. This report aims to assist in this task. Detailed comments are listed below, I hope they are clear and useful.

We addressed all comments/suggestions by the reviewer, please see our responses below.

Title and abstract

Title and abstract suitably reflect the content, while there are few minor deficits in the abstract:

- It is stated that 'More than 80% of the EU agricultural imports will become highly vulnerable to drought in future because of climate change'. Paper doesn't provide background for this statement anywhere and it seems to be contradictory to the results as presented in rows 330-332. It should be clarified in the introduction section if this percentage refers to earlier research, or in the results/conclusions if it refers to the results of this study.

We corrected it, it should be 44%, as indicated in lines 330-332 (365 in the revised text).

Introduction

The introduction section covers the paper and presents sufficient background and justification for the paper, while lacks references to some of the claims made. This is a shortcoming of the paper that repeats itself in the discussion section more significantly. row 4: Reference 1 on the state of European agriculture is from a newspaper. It would be recommendable to find the original sources of the news article and refer to those instead for more scientifically convincing argument.

We changed the first reference to a scientifically published article.

row 5: Reference 2 on the 'climate change forecasts' is similarly a secondary reference and the argumentation would benefit if the original sources would be referred to instead.

We think this reference is a valid source, an official publication by the EU Commission. We referred to this source since it presents a comprehensive summary of various publications about both direct and indirect (via trade) impacts of climate change to the EU's agriculture. Therefore, we keep as it is, we hope the reviewer also agrees our logic behind our decision.

Sentence in rows 12-15 is obviously true while not an original argument and reference to the earlier studies proving this should be presented.

We added a reference describing this dependency (new reference no:9).

Moreover, while this is not the foci of this paper, it could be considered that it is not directly the deficit in soybean production per se that poses the risk but the animal production sector's dependency on soybean which could be replaced by other feed plants that are more easily available and producible in the European context (see e.g. Sandström et al 2018).

We modified the sentence in row 9 to explicitly mention relevance to this topic (and included new references, no 9). Furthermore, we added a sentence about role of "replacement" in the discussion and conclusion sections (please see lines 386-388 and new references 9,11,12,13).

The claim presented in rows 19-21 is interesting and somewhat dramatic and it partly justifies this study, while, it would be more convincing if presented with a reference.

Three references (no: 14,15,16) were added at the end of the same paragraph, which show potential impacts that can happen under such events (for example, feed prices as happened in the USA).

In rows 23-25 authors refer to previous ('past decade') scientific literature on sectoral vulnerability assessments. The argument could be stronger if the literature that is referred to here would be defined more precisely (the references are mainly about integrated assessments).

A sentence was added to briefly described the nature of such applications, please see lines 29-32, in the revised manuscript.

Moreover, the EEA reference does not necessarily fit to the presented description of this literature (scientific).

This reference, even though not a peer-reviewed publication, nicely describes the cross-border climate relevance and a summary of previous studies. We think it is an important publication

since it is published by an agency of the EU working on this topic. Consequently, we would like to keep this as a reference in this section.

There are also more recent studies that present analyses of broader than country-scale sectoral vulnerabilities to cross-boarder impacts (see e.g. Groundstroem et al. 2019).

Two new references are added, see references no 9 and 28.

rows 25-29: A direct quote from the IPCC is presented. This should include an appropriate reference of source.

A reference is added (see, no:22).

The final section of introduction could benefit from introducing references in rows 41-52; 47-49 to clarify the original sources of the used concepts (CCVS, green virtual water, SSP2) and the scenarios. It is now unclear whether and which of these are the authors' own and which from other sources.

References related to green virtual water and SSP were included (references with no 35 and 36). CCVS is the term we formulated for the cross-border climate vulnerabilities for the research presented in this paper.

Illustrations

Illustrations of this paper are necessary and relate to the text, while it is not clearly stated why only the 2050 RCP6.0 example is chosen to be presented in fig. 2 and fig. 5. My understanding is that these exemplify the most significant findings. However, to make this section more convincing, the complete results could be made available e.g. in supplementary material.

We, indeed, used only RCP 6.0 results in illustrations to exemplify the most significant findings. We added one figure about RCP 2.6 in the newly introduced SI section.

- **Figure S.1:** Cross-border climate vulnerability score of the EU's agri-food economy to drought per exporting country in 2050 under the RCP 2.6 concentration pathway.

Even though the CCVSs in each cell are not same for the RCP 6.0 and 2.6 scenarios, when we convert them to the vulnerability levels (as done in Figure 5), the maps for each scenario are almost identical. This is because many of the values in each cell fall between under the same range/category. For example, CCVS value of 0.8 and 0.9 fall under the same category of "low", thus represented under the same color identification in the maps. If we added RCP 2.6 version of the Figure 5, it would look exactly the same. For this reason, we only provided for RCP 6.0. To illustrate what we mean, we here put vulnerability level maps for soybean both under RCP 2.6 and RCP 6.0

Discussion

The paper presents a linear and quantified assessment of an element/aspect of a highly complex socio-environmental phenomenon with temporally far-reaching scope. This inevitably involves major trade-offs with the depth of the assessment and this is duly stressed by the authors as major limitations of the current study that should be addressed in future studies (rows 306-318). However, the discussion section very scarcely presents references of such research that the findings contribute to (see detailed comments below).

We added these references (see below). We also included a paragraph in the discussion section regarding how our approach can contribute to overall goal of sustainable development and system thinking, please see lines 349-356.

Also, since authors only here clarify that these limitations may effect the outcomes of the study, this point should be presented earlier with short notion (while highlighting that these results are a first step in this important next level of understanding the broader-crossing vulnerabilities).

We added a sentence referring to this, please see lines 332-333.

The crops assessed in the study are based on that they are currently imported in volume and they are dependent on the EU's external rainfall dependency (row 136). Some of these are such that may be replaced by other products (e.g. soybean with broad bean) (see e.g. Sandsröm et al 2018). This is addressed briefly in discussion (row 266). To relate the discussion more strongly to relevant on-going scientific discussions literature on transformative adaptation might be found interesting.

We added such references and a sentence related to alternative sources for soy, please see lines 286-289. We added relevant references, please see reference no: 11,12,13

Conclusions

row 330: authors state ' We found that more than 44% of the EU's agricultural imports will become highly vulnerable to drought in future because of climate change.'. Similar to the percentage (80%) presented in the abstract, it is unclear how this number is drawn (is it an average of the proportion of all six scenarios assessed and for the crops assessed in this study, or another method, or from another source?).

We corrected the abstract, to 44%, which is drawn from Figure 3.

Other minor comments

The paper would benefit from language check by an expert. Some challenging formulations of sentences make it hard to grasp the meaning of the text at points, for example sentences in rows 29-31; 43-45; 193-194. In sentence in rows 29-31, for example, it is not clear which studies are focused on the 'consumer perspective' and how does this relate to the cross-boarder aspect?

Row 29-31: "consumer" was wrongly placed, the sentence is corrected. (new 35-36)

Row 43-45: revised (new 47-49)

Row 193-194: revised (new 210-211)

Language check has been done.

While I like how section 'Cross-border climate vulnerability of the EU: key-imported crops' is written, the presentation of results of the different crops appear somewhat inconstant leaving an unfinished impression. E.g. the term presented in row 161: 'key player in the cocoa sector' would benefit from specifying what is meant by this, particularly for the less economy-wise reader. The following paragraph on coffee-related results is more easy to grasp. Similar to how societal change (in values/preferences that affect the demand) is speculated to impact on sugar cane (rows 218 – 220), there is literature on similar processes related to e.g. how animal product demand might change and thus reflect on maize and soybean. row 262: I suggest not use term mitigate in relation to climate risks. It is easily confused with climate change mitigation and thus doesn't benefit the papers clarity to wider audience.

Row 161: we deleted "key" and replaced it by "important".

Row 218-220: we deleted the sentence as it is not substantiated sounded more like a speculative text.

Row 262: we changed the word "mitigate" to "prevent"

Methods

Research methods seem to be appropriate and explained clearly enough in the Method and Data section. As the quantitative methods used in this paper are not my field of expertise, I would like to present a comment from the perspective a less experienced quanti-method reader:

The temporally far-reaching results that the study presents are much needed in picturing the potential futures that such large economies as EU and it's vital sectors such as agriculture need to be prepared for. These results, however, inevitably include high level of uncertainty – a reason why earlier studies have claimed such assessments impossible referring to e.g. the uncertainty in crop water requirement changes due to productivity increase (see e.g. Sandström et al. 2018). It remains unclear how these uncertainties are addressed in this paper for an unexperienced reader of the methodology.

Data used in this paper come from peer reviewed scientific literature and have been accessed from publicly available global databases. In many cases, models have been used to produce the data. These input data, thus, include uncertainties inherent in datasets produced.

Uncertainties affecting this work can be split in two groups; the first one relates to those uncertainties that origin from the water footprint calculations (water embedded in the production), and the other relates to the drought severity calculations (both under current and future conditions). We acknowledge these uncertainties while at the same time tried to minimize them by

using various GHMs, GCMs, scenario's, etc. using the multi-model and multi-scenario approach (Warzawski et al., 2014; Piontek et al. (2014); Prudhomme et al.; 2014)

We'd like to stress here that we looked at relative differences only, i.e., the difference between current and future. So, the drought severity was calculated for current and future conditions based on the same multi-model ensemble, and then were compared (difference between future and current). Our results are primarily based on the combination of green water use estimates (specified as constant) and drought severity estimates for the different periods. The same uncertainties apply for both sides of the comparison, so they become less relevant. Being the outputs of the same models, both current and future condition, and inheriting the same assumptions and uncertainties allowed us to make comparisons as described in our article.

References: please see the special issue in PNAS, https://www.pnas.org/global_climate

Reporting summary file seems to be sufficient and correct.

REFERENCES

Groundstroem, F. and Juhola, S., 2019. A framework for identifying cross-border impacts of climate change on the energy sector. *Environment Systems and Decisions*, 39(1), pp.3-15.

Sandström, V., Lehtikoinen, E. and Peltonen-Sainio, P., 2018. Replacing imports of crop based commodities by domestic production in Finland: potential to reduce virtual water imports. *Frontiers in sustainable food systems*, 2, p.67.

Reviewer #3 (Remarks to the Author):

The paper presented climate change vulnerability of different EU imports based on exposure and sensitivity of drought in regions of production. The methodology of using green virtual water imports as a base to determine vulnerability of EU crop imports is both a novel and interesting attempt. Most of the earlier studies focused on direct impact of climate change on crop yields and production. The paper concluded that some of the EU crop imports such as cocoa, coffee and sugarcane will be highly vulnerable to drought in the exporting country and suggested that it will have implications on EU agricultural economy. This paper is relevant and can be very useful to academics, researchers and policy makers working on agriculture economy, international trade and food security. It has potential to generate good discussions on these topics. The use of green virtual water volume to examine the impacts of climate change on EU crop imports can also be used to measure sustainable agricultural imports to the EU. There, however, are few issues in the methodology and analysis that I believe needs to be addressed to improve the paper for publication.

We would like to thank to the reviewer for her/his positive and constructive comments. We addressed each comment, please see the relevant part for our responses.

The main issue I am concerned with is the way impact of climate change crops is addressed in this paper. The authors based their analysis in examining vulnerability of crop imports over drought events in exporting regions considering only the impact of drought on quantities of crop produced (represented by virtual green water volume) and exported.

Crop production under climate change are not affected by soil water contents (in case of drought) alone but also by several other factors such as length of growing season, temperature and potential CO₂ fertilisation etc. There are many studies which showed that crop biomass accumulation (including water content) is highly sensitive to some of these factors (Tubiello et al, 2000; Donatelli et al 2015; You et al, 2009; Challinor et al, 2014). The effects of these factors on crop production are mostly manifested together, synergistically sometimes, so the impacts of climate change would be misleading if effect of only one factor is isolated and used in an analysis.

We fully agree with this comment of the reviewer. Climate change impacts cannot just be simplified as a single "soil moisture anomaly" issue. As mentioned by the reviewer, multi-stressor analysis would be more comprehensive to address these vulnerabilities. We mentioned this limitation clearly in the discussion section (please see lines 332-339). We purposely used a single stressor because of two main reasons: (i) we wanted to merge green water use (and virtual water imports) with climate vulnerability assessment and soil moisture anomaly was a proxy to green water scarcity (ii) multi-stressor analysis would have required more specific input data per crop type. Since we included all crops imported in the analysis, such input data would not be available. For example, it would not be possible to do such analysis for perennial crops such as cocoa, coffee globally. More importantly, we wanted to focus this analysis on the sensitivity of the system to drought. Including other stressors would have made it difficult to understand what the vulnerability to drought is on its own.

Another, somewhat similar, issue is on adaptations under climate change. Farmers are known to make small adjustments to mitigate any adverse production conditions. It is argued by many researchers that a climate impact study without a provision of potential adaptation will be highly exaggerated (Medelsohn et al, 1996; Tan and Shibasaki, 2003; Wreford et al, 2010 etc). Will the results in this paper still show a higher vulnerability to some crop imports, for instance, if the green virtual water volume is decreased but there is an increased use of irrigated water to minimise loss in

production? IPCC definition of vulnerability, as stated by the authors on line 27-29, includes adaptive capacity as one of the essential functions to measure vulnerability. That is why many climate change studies included potential adaptations in the analysis of climate change impacts on agriculture (Wreford and Adgar, 2015; Donatelli et al, 2015 etc.).

The authors acknowledged that climate change factors other than drought effects and adaptations to minimise those effects are not included in the paper. Nevertheless, in their conclusions, the authors suggested the results and analysis presented in this paper would be useful to the EU policy makers developing climate change strategies and trade bilateral relationships. I believe to make the results and analysis presented in the paper stronger and more justifiable, the authors need to adjust the climate change vulnerable scores to account overall climate change impacts and potential adaptations on crop production the exporting regions.

We fully agree with the comments of the reviewer about adaptation capacity. To reflect this limitation, we revised our method, included climate change adaptive capacity of agriculture in each exporting country in our models and calculations. We re-calculated CCVSs and added relevant text to the manuscript. Relevant sections can be found in the revised manuscript

Our goal in this research to look at particularly drought severity component as the potential hazard and combine it with green water use (green virtual water imports) as sensitivity. So, we looked at green water use and compared to green water availability (or anomaly). Incorporating other stressors may not correspond well with green water use sensitivities. Thus, we did not further revise the study by incorporating multiple stressors.

The main theme of the paper is exploring the impacts of future drought events under climate change. However, there is not much review on past drought incidents and their impacts on the EU agriculture much except for a media report at the start of the introduction. There are plenty of reports and studies that shows the severity, intensity and impacts of droughts in different years in the EU (Spinoni et al., 2016; Ciscar et al 2014; Wreford and Adger, 2011; Benzie et al., 2011; Garcia-Herrera et al 2010 etc.) and elsewhere (Qureshi et al 2013; Medellin-Azura et al., 2016). A thorough review of some of these studies will provide a good insight to the objectives of the paper.

Even though our focus is on drought vulnerabilities under climate change, we specifically address cross—border side of it. Since our focus is not within the borders of the EU, we did not provide extensively reviewed publications related to such local events. We think providing such a thorough review of some of these studies will take the focus on those studies and would be disproportional to the cross-border analysis done.

However, we understand from the comment that further references are needed to strengthen the introduction section. Therefore, we revised the manuscript as follows:

- We changed the first reference to a scientific publication.
- We added further references in addition to the first reference as suggested by the reviewer (references no 2,3,4,5,6,7).

The statement on line 29-30 indicates that the current paper is looking at climate vulnerability through consumer's perspective. However, I feel that the analysis presented in this paper is still production based as it uses CCVSs that only includes virtual water volume (in other words quantity of production). A consumer's perspective will include availability (supply), accessibility (potential

Line 43 – CCVS is stated for the first time here. The term is defined later in the text (line 58) but providing a definition when mentioned for the first time will be easier for a reader.

We added explanation at this section.

Line 56 – I found it a strange place to state the main conclusion here in the main text before providing results and analysis.

We removed conclusion word from that sentence.

Line 72: 'They will remain at similar levels in 2050..'. Please check as I see a decrease in CCVSs in 2050 in Figure 1.

We added "slightly decrease"., next to that sentence.

Line 115: The authors stated only 4 levels of drought vulnerabilities; low, medium, high and extremely high. The levels are different from what is indicated in Figure 3.

- corrected

Similar to Figure 1, Figure 3 also shows that drought severity decreases in 2085 compared to 2030 and 2050. This is contradicting the fact that climate change impacts will increase further especially under RCP 6.0 in future. Do the drought severity decrease over time? A reference to drought severity map would be helpful to see if this is correct.

We calculated drought severity ourselves, using soil moisture levels of multi-model assembly outputs from global hydrological models. Thus, the drought severity maps are not from another source. However, our results are in parallel with the results of other studies related changes in soil moisture. For example, Lu et al., provided a map of soil moisture changes for all RCPs and compared to current levels for 2071-2100 (represented as 2085 in our study). Their maps (e.g., Figure 1 of their study), shows relative increase in soil moisture in RCP 6.0 compared to current conditions, in some parts of India, Pakistan, sub-Saharan Africa and Brazil etc., which are exporting regions to the EU (for 2085). Consequently, their result supports the results of our study in terms of increased wetness in 2085 for some regions.

Reference:

Lu, J., Carbone, G.J. & Grego, J.M. Uncertainty and hotspots in 21st century projections of agricultural drought from CMIP5 models. *Sci Rep* 9, 4922 (2019). <https://doi.org/10.1038/s41598-019-41196-z>

Line 136: This statement is a repeat of an earlier statement.

We removed this statement.

Line 162 – Is it a typo? - chocolate industry?

Corrected.

Line 211: Does this mean 87% of olive imports are irrigated production?

No, it means 13% of the total green water use related to olive products demand is external, 87% is internal.

Line 260: In other climate change studies, the climate change scenario years for example, 2030 uses 30 years averages and represents years in the decade 2030s. Do the scenario years in this paper represented similarly? There is some indication of that in discussion, but I guess the authors need to describe the scenarios in some more detail in the methodology.

Yes, it is as the reviewer described. We mentioned this in the method ,please see the lines 549-550.

Line 261: Sugarcane? Which I guess is the most vulnerable.

Sugar cane is added.

Line 279-280: There is clearly differences in vulnerability between different crops and between different regions. However, there is no attempt to show why is that? Is that because some crops need less water for production, or it is because a crop is produced mostly on irrigated water in some regions which lowers its vulnerability.

We showed such differences in vulnerability in Figure 5, per crop, per region. These lines are in the discussion section. We chose not to repeat the results given in the previous section in this part. We provided some examples in the lines 300-307.

Line 331: Conclusion of 44% vulnerable was not stated anywhere in the Results section.

This number is taken from Figure 3. RCP 6.0, summation of high and extremely high is equal to 44% of the imports (25% + 19%).

Line 331: a typo - EU"s

Corrected.

Line 333: Increase in drought severity by 35% not mentioned anywhere in the text except in the abstract and conclusion?

It is taken from Figure 1, which shows CCVs is around 1.35, which means it increased by 35%. This was explained in the text in the relevant section (see lines 65-67).

Line 339: Production in the vulnerable regions would be disrupted due to drought. This may not truly disrupt supplies to the EU as the EU imports may shift to other sources. This again shows a weakness in not adjusting for adaptations under climate change scenarios in this study.

We included adaptive capacity in our study. We re-calculated CCVs, please see the revised version.

Line 364 – It is stated here that demand production scenarios are based on a number of drivers including population growth. Climate vulnerability for most of the commodities are decreased in 2085 which is due to reduction in population in the EU. But as the population in most of the exporting countries most probably will increase in future, don't they have any impact on CCVs? On

line 369, it is suggested that water footprint for outside EU is assumed to be unchanged. How it is possible to keep green water footprint outside the EU constant when the impact of drought is examined on production in those regions? Surely, drought events will change water footprints in for the productions in those regions.

We agree that the internal demand for the commodities by the exporting countries will change (larger or less demand) and this may be impactful in trade structure. This subject is one of our limitations. We indicated this limitation in the article in the discussion section, please see lines 340-342.

Keeping green water footprint of crops, soil moisture uses per tonne of crop produced, is a simplification of our analysis. Calculating green water footprints under climate change needs two type of information: crop yield changes and evapotranspiration changes. The first, as we described in our other responses, was not possible due to the large list of crops and global scale of the analysis (e.g., cocoa, coffee etc). Since it was not possible to model yields for all imported per grid-based location, we used soil moisture anomaly as a proxy to potential yield changes i.e. green water scarcity. Therefore, we kept the green water footprints constant. We acknowledge that WFs will not be same in future and this is a limitation of the study. However, as our study is a vulnerability, not an impact, we think current levels of green water use gives us a good indication of sensitivity levels for comparison purposes among various crop types and locations.

Line 408: In the methodology, sensitivity is defined as quantity of water imports. For me sensitivity means the responsiveness to a change so wonder how sensitivity was determined?

Sensitivity to drought severity is about the scale of green water use. More green water is needed for a location, more it is sensitive to soil moisture anomaly. For example, if green water use is zero, production would not be impacted by soil moisture anomaly. Thus, green water use size is a proxy to sensitivity to drought changes in our analysis.

Line 482 – The expected production is determined by using EU’s share to world production. Why this method is better than using historical trend adjusted to population growth (as some of the sectoral models such as CAPRI and FAPRI use to project)?

The method we used is based on global balance between demand and supply of individual crops/crop groups (thus volumetric balance). It is not better or worse than other approaches, but a different one. This approach used in previous studies of us (published), therefore we used the same supply-demand balance model we have previously developed.

Line 494: To calculate drought severity, the authors assumed a threshold value $q_0(\theta)$ to be 20%. I am not sure what this 20% means, is it the soil moisture level 20% of normal soil moisture or something else?

We followed the method described by Sheffield and Wood (2008) for calculation of the drought severity. They describe the $q(\theta)$ a percentage, less than a chosen threshold, $q_0(\theta)$. In this study we used a value of 20%, which reflects conditions that occur only once every 5 years for a particular month on average and so reflects medium and rare events and can be applicable to identifying historic events at global scales. Sheffield and Wood applied 10% for this ratio, since they were interested in extreme events (happens in every 10 years). We hope this explanation is sufficient for the reviewer.

Equation 15: what does the ratio $S_{p,y,i}/S_{p,y,e}$ represent? Why it is needed when it uses summation over $n=i$?

This ratio was introduced to distribute demand changes from exporting country to grid cell base in each country. The demand change is at country level and drought severity is at grid level. To calculate vulnerability level at grid scale, we had to go from country level to grid-scale. This ratio is this distribution factor. We added this explanation to the relevant part in the method (after the equation).

While calculating severity and exposure, the authors used 2010 data as base year representing today. This is year 2020, why the authors used base data 10 years old and not use more current data year?

This is related to temporal scale used at hydrological models used in the multi-model ensemble. We used their modeling for current scale of drought severity. And most of those models do not provide data beyond 2010.

Table 2: $0.5 < V \leq 1$ is stated as 'lower' level. Is it a 'low' level as defined in the text?

It is low, we corrected.

Reference 17: title missing

Added.

Figure 5: In line with adding the RCP 2.6 pathway map per importing country to the supplements, it would seem logical to also present the RCP 2.6 pathway maps per crop in the supplements as well.

Supplementary material: Figure S.1 lacks color coding of the index scale.

Looking forward to reading the published paper,
With best regards,
Janina Käyhkö

Reviewer #3 (Remarks to the Author):

I am happy with the revised texts and replies to the issues I raised in my review. I acknowledge the authors to address my comments in such a thorough way.

Respond to the reviewers

Cross-border climate vulnerabilities of the European Union to drought

Ertug Ercin ^{a,b}, Ted I. E. Veldkamp^c, Johannes Hunink^d

a R2Water Research and Consultancy, Amsterdam, the Netherlands

b Institute for Environmental Studies, Vrije Universiteit Amsterdam, the Netherlands

c Amsterdam University of Applied Science, Amsterdam, the Netherlands

d FutureWater, Cartagena, Spain

*Correspondence: ercin@r2water.nl; Tel.: +31617366 472

All responds are written in “red color”.

Dear authors,

I was pleased to read your revised manuscript and see that you have addressed in detail all comments made earlier. This has made a significant positive impact on the paper. Below I present minor comments that I hope to assist you in finalizing the paper. I recommend the paper to be accepted for publication with minor revisions.

Abstract, line 26-27: Please, clarify whether imports from the given countries will be less vulnerable than previously or in comparison to the previously mentioned countries.

Clarified.

lines 64/65: an extra parenthesis

Corrected.

line 94: Please, clarify in the first sentence already that you refer to temporal change i.e. “will be more vulnerable to drought in future” or “The impacts of climate change in future will increase the vulnerability” or similar.

First sentence is revised.

94-100: It is hard to grasp what is the temporal baseline of the assessment. One needs to go to the methods section to search this information (and find that the baseline years are a bit different for the different variables e.g. 2010 for sensitivity, 2006 for severity, 2018 for adaptive capacity, and that the baseline year for CCVS is 2010). If possible, please, explain or make a reference to the methods section or a footnote that the baseline for CCVS score is 2010 and/or that it represents the current moment/today.

Clarification is added to indicate that the baseline is current situation.

131: Wording "equip agriculture areas" is unclear. Do you mean "agricultural areas equipped with ..."?

Corrected

133: Unclear what is meant by "will be one, 0,2 higher". Do you mean "1,2 higher" or "one which is 0,2 higher"?

Corrected.

156: Sentence lacks a word ("however, they do not..").

Corrected.

163-168: To clarify the text, please, be coherent when referring to the proportion of imports from low/low-medium drought severity areas currently and in future i.e. either refer to low or to low/low-medium in both sentences.

The sentence is corrected to make it more coherent.

181: To clarify the text, please, be consistent in use of the severity level terms (very high=extremely high?).

Consistency is checked and corrected where necessary.

194: This is not the first time you refer to USA so perhas introduce the abbreviation earlier (in line 133) if necessary.

Added in line 141.

406: unnecessary italics in "change"?

Corrected.

524: Spacing between rows lacking.

Corrected.

764: reference missing

Corrected.

Figure 1. It is laborous to pick out the difference between RCP 6.0 and RCP 2.6 results and the temporal development simultaneously as the color coding between the two concentration pathways is the same. Is there a reason why Fig.1 and Fig.4 don't use the same color coding? If so, in lack of better solutions, I suggest to tone down the other

pathway color palette slightly. Also, please, clean the appearance of the figure regarding the y-axis lower levels which are now overlapping with the captions.

They are same color since they refer to the same components, "socio-economic" "hydro-meteorological factors" and "combined". Making them different colors need additional explanation of the same variable in a different color. Therefore, we would like to keep as it is. Also, Figure 1 and 4 organized differently. This comment would be valid if we design the figures under same structure.

Figure 4 (same comment as Fig.1): Please, clean the appearance of the figure regarding the y-axis lower levels which are now overlapping with the captions.

Corrected.

Figure 5: In line with adding the RCP 2.6 pathway map per importing country to the supplements, it would seem logical to also present the RCP 2.6 pathway maps per crop in the supplements as well.

This comment was raised previously, and we responded as follows:

"Even though the CCVSs in each cell are not same for the RCP 6.0 and 2.6 scenarios, when we convert them to the vulnerability levels (as done in Figure 5), the maps for each scenario are almost identical. This is because many of the values in each cell fall between under the same range/category. For example, CCVS value of 0.8 and 0.9 fall under the same category of "low", thus represented under the same color identification in the maps. If we added RCP 2.6 version of the Figure 5, it would look exactly the same. For this reason, we only provided for RCP 6.0. To illustrate what we mean, we here put vulnerability level maps for soybean both under RCP 2.6 and RCP 6.0"

Supplementary material: Figure S.1 lacks color coding of the index scale.

It does, on the top of the figure.

Reviewer #3 (Remarks to the Author):

I am happy with the revised texts and replies to the issues I raised in my review. I acknowledge the authors to address my comments in such a thorough way.